

# Mir-421 in plasma as a potential diagnostic biomarker for precancerous gastric lesions and early gastric cancer

Jianlin Chen[1,*], Lihua Wu[2,*], Yifan Sun[1], Qi Yin[1], Xianhua Chen[1], Siqun Liang[1], Qingyan Meng[1], Haihua Long[2], Fangying Li[2], Changjun Luo[3] and Xiaorong Xiao[4]

[1] Department of Clinical Laboratory, Affiliated Liutie Central Hospital of Guangxi Medical University, Liuzhou, Guangxi, China

[2] Department of Digestive Internal Medicine, Affiliated Liutie Central Hospital of Guangxi Medical University, Liuzhou, Guangxi, China

[3] Department of Internal Medicine-Cardiovascular, Affiliated Liutie Central Hospital of Guangxi Medical University, Liuzhou, Guangxi, China

[4] Department of Science and Education, Affiliated Liutie Central Hospital of Guangxi Medical University, Liuzhou, Guangxi, China

* These authors contributed equally to this work.

Corresponding author
Yifan Sun, sunyifan13@126.com

## ABSTRACT

**Objective**. MicroRNA (miR)-421 plays a key role in cancer progression. It has been reported that circulating miR-421may be a potential tumor marker for the diagnosis of several cancers. However, the role of miR-421 in plasma as a potential biomarker in the diagnosis of precancerous gastric lesions (Pre) and early-stage gastric cancer (GC) remains poorly understood. In this study, we investigated miR-421 in plasma as a novel potential biomarker for the detection of precancerous gastric lesions and early-stage (GC).

**Materials & Methods**. The miRNA content was determined by quantitative real-time polymerase chain reaction (qRT-PCR). MiR-421 content in all subjects was normalized by endogenous miRNA (miR-16). The diagnostic value of miR-421 for Pre and GC was assessed by comparing receiver operating characteristic (ROC) analysis with traditional tumor markers, including CEA, CA125, CA153, CA211 and CA50. The correlation between the expression of miR-421 and the pathological characteristics of Pre and GC was analyzed.

**Results**. Elevated expression of miR-421 in plasma can robustly distinguish the normal population from Pre and GC cases, especially in the early stages of gastric cancer cases (all $p < 0.05$). The ROC analyses showed that the area under the ROC curve (AUC), sensitivity, accuracy and Youden index of miR-421 were superior to traditional tumor markers (CEA, CA125, CA153, CA211, and CA50) in GC diagnosis, while its specificity was higher than CEA, CA153 and CA50 (all $p < 0.05$). MiR-421 in plasma had higher AUC value than AFP, CA153, CA211 and CA50 in the diagnosis of Pre (all $p < 0.05$), while specificity, accuracy and Youden index of miR-421 was only lower than CA211. The efficiency of miR-421 in the diagnosis of GC was significantly higher than that of CA211 and CA50, and it was significantly higher than CA153, CA211 and CA50 in the diagnosis of Pre (all $p < 0.05$). In addition, up-regulation of miR-421 occurred initially in precancerous gastric lesions as well as in the early stage of GC.

**Conclusions**. Overexpression of plasma miR-421 is a novel biomarker for the detection of precancerous lesions and early gastric cancer.

## INTRODUCTION

Gastric cancer (GC) is one of the most common malignant tumors in China (*Zheng et al., 2019*). Due to the lack of tumor markers and specific symptoms and signs, most tumors have metastasized by the time of diagnosis. Studies have shown that patients with advanced GC have a poor prognosis with a 5-year survival rate of less than 25% (*Chan, Wong & Lam, 2001*). However, if intervention and treatment are performed in the pre- or early stages of gastric cancer, the five-year survival rate will rise to 95% (*Craanen et al., 1991*). Therefore, it is very important to detect and treat gastric cancer at an early stage.

Gastric cancer originates from the malignant and aggressive proliferation of gastric mucosal cells. Most are developed and evolved by the chronic inflammation of gastric mucosa under the influence of multiple factors, mainly in four stages (chronic superficial gastritis → chronic atrophic gastritis → intestinal metaplasia and dysplasia → gastric cancer). Chronic atrophic gastritis is currently recognized as a precancerous disease, intestinal metaplasia and dysplasia is a precancerous lesion closely related to the development of gastric cancer. Currently, Gastroscopy combined with biopsy is the gold standard for diagnosis of gastric cancer. However, it may not be appropriate to carry out large-scale screening and early detection in most countries, including China, due to acceptability, availability, cost and other reasons. GC related tumor markers such as, cancer embryo antigen (CEA), pepsinogen (PG), carbohydrate antigen 199 (CA199) and carbohydrate antigen 724 (CA724), gastrin-17 (G17) have been used clinically for many years, but many do not have sufficient sensitivity and specificity for GC screening (*He et al., 2013*; *Pectasides et al., 1997*; *Yang et al., 2014*). Studies on their role in monitoring early gastric cancer have rarely been reported. Therefore, there is a need for reliable and non-invasive biomarkers for early detection and mass screening of GC.

MicroRNAs (miRNAs) are a class of endogenous non-coding RNA molecules (length of 18–25 nucleotides) that exert their roles by base pairing between the seed region of miRNA or 3′-un-translated regions (3′-UTR) of the target gene (*Lee & Ambros, 2001*). Studies have confirmed that miRNAs' expression are frequently de-regulated in human tumors (*Hernando, 2007*; *Lu et al., 2005*). Recently, circulating miRNAs (miRNAs present in blood) have attracted attention due to their stability and ease of measurement. Many circulating miRNAs have been quantified and identified by qRT-PCR as tumor markers to detect GC, such as miR-223, miR-21, miR-218 (*Li et al., 2012*; *Zhou et al., 2015*), and miR-18a (*Tsujiura et al., 2015*) in plasma, and miR-20a (*Yang et al., 2017*) in serum. However, there is limited research into circulating miRNA markers for the diagnosis of precancerous gastric lesions and early-stage gastric cancer, and most of these studies have not compared the value of these miRNAs with traditional tumor markers in the diagnosis of gastric cancer and precancerous lesions.

MicroRNA-421 (miR-421) is a molecule that has been studied extensively. It has been investigated and found to be expressed aberrantly in various types of cancer (*Li et al., 2018*; *Wang, Liu & Shen, 2018*; *Zhou et al., 2016*). Previous studies showed that miR-421 rises notably in tissue and gastric juice, as well as in blood mononuclear cells and serum of gastric cancer (*Liu et al., 2015*; *Zhang et al., 2012a*; *Zhao et al., 2015*), suggesting that miR-421 seems to be an ideal biomarker for gastric cancer. However, most of these studies failed to point out at which stage in the evolution of gastric cancer miR-421 can be detected in the peripheral blood, and an optimal circulating miRNA should have the ability to distinguish different phases in the course of cancer development. Some critical issues remain unresolved. Firstly, the expression of miR-421 in the plasma of precancerous patients has not been fully investigated. More importantly, the value and significance of miR-421 in the early diagnosis of gastric cancer are still not clearly stated. The present study aimed to clarify the diagnostic value of miR-421 in the early stage of gastric cancer and in the precancerous lesions of gastric cancer and to explore its relation with the progression of GC.

## MATERIALS & METHODS

### Clinical specimens

Plasma samples from 90 GC patients, 89 patients with precancerous (Pre) lesions and 45 normal healthy controls (NC) were collected at Affiliated Liutie Central Hospital of Guangxi Medical University in Guangxi province, China and the levels of miR-421 were screened by qRT-PCR. GC patients were confirmed by gastroscopy combined with biopsy. Tumor type and stage were identified according to the Union of International Cancer Control (UICC) tumor-node-metastasis (TNM) system, 7th edition. The histology of all patients was evaluated according to the World Health Organization (WHO) criteria. Patients who had received Radiation therapy or chemotherapy were excluded from the study. The pathological outcomes of GC and Pre patients were obtained by two experienced pathologists. In addition, 45 healthy people without a previous history of diabetes, heart disease, hypertension, and cancer were chosen as normal controls (NC). We compared the plasma levels of miR-421 in patients with gastric cancer, and precancerous lesions, and in normal controls to assess the feasibility of miR-421 as a novel non-invasive biomarker for early GC detection. The expression levels of conventional tumor markers in GC, Pre, and normal controls were analyzed to evaluate the specificity and sensitivity of miR-421 for early diagnosis of gastric cancer. This study was subject to approval by the Ethics Committee of the Affiliated Liutie Central Hospital of Guangxi Medical University and written informed consent was issued by all study participants.

### Plasma preparation and storage

8~10 ml of venous blood of each participant was collected using ethylenediaminetetraacetic acid (EDTA) anticoagulative tubes immediately. Cell-free plasma was separated within 2 h after collection by 3,000 rpm,5min centrifugation to prevent contamination by cellular nucleic acids. Plasma transferred into RNase/DNase-free tubes was stored at $-80\,°C$ until

MicroRNA extraction. Plasma samples for conventional tumor markers and *Helicobacter pylori* antibody determination were separated and kept at $-20\,^{\circ}C$ until assayed.

## MicroRNA extraction and qRT-PCR assays

MicroRNAs were isolated from 200 μl plasma utilizing Blood (Serum/Plasma) MicroRNA Extraction and Purification Kit (spin column) (Novland Co., Ltd, Shanghai, China) following the manufacturer's protocol. The concentrations of miRNA from plasma samples were quantified using a NanoQ micro-volume Spectrophotometer (CapitalBio, Beijing, China). The initial template for qRT-PCR was 2ul. Circulating miRNAs expression was determined using one step Stemaim-it miR qRT-PCR Kit Quantitation (Taqman probes) (Novland Co., Ltd, Shanghai, China), while miR-16 served as a reference miRNA. Expression levels of target miRNAs were performed on the ABI-7500 PCR system and calculated by the cycle threshold (Ct) values with SDS 2.0 software (Applied Biosystems, Foster City, CA, USA). Samples with Ct values greater than 30 were excluded. The levels of miR-421 in different samples were calculated by the $2^{-\Delta Ct}$ method, in which $\Delta Ct = Ct$ (miR-421)–Ct (miR-16).

## Conventional tumor markers and Helicobacter pylori antibodies

Conventional tumor markers were tested by electrochemiluminescence immunoassay according to the standard procedure of Roche Company's kit and Roche E170 automatic immunity analyzer. *Helicobacter pylori* antibodies were assayed following the Anti-helicobacter pylori antibody detection kit (ELISA) (Beier Biological Engineering Co. Ltd, Beijing, China).

## Statistical analysis

All data were analyzed using MedCalc statistical software (v18.2.1) and GraphPad Prism 7.0. The levels of miR-421and conventional tumor markers in plasma among Pre, GC patients, and health cases were established by the Mann Whitney test. Receiver operating characteristic curves (ROC) and the AUC were constructed to evaluate the diagnostic values of each tumor markers. The optimal cut-off thresholds were determined by using the highest Youden index. The Chi-squared test was utilized to analyze the associations between the expression level of miR-421 and the clinicopathological factors. All differences were examined statistically significant at $p < 0.05$.

## RESULTS

### Expression of miR-421 and tumor markers in plasma of GC patients, Pre patients, and normal controls

This study recruited 90 GC patients, 89 Pre patients, and 45 normal healthy subjects as controls. Plasma miR-421 levels were detected by qRT-PCR and analyzed using miR-16 as a reference miRNA. To verify whether miR-16 was adaptable to be a reference miRNA in our system, the expression levels of miR-16 were detected. As shown in Fig. 1A, there was no statistical difference in the expression of miR-16 between the normal and gastric cancer group, or the normal and precancerous group, which implied that miR-16 was a suitable reference control for detecting miR-421 in plasma samples. The qRT-PCR results showed
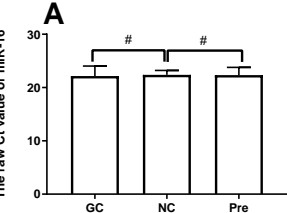 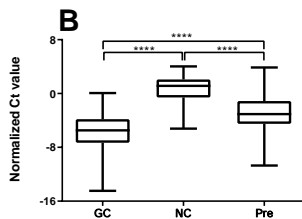 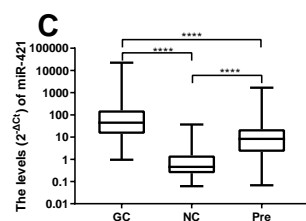

**Figure 1** MiR-421 expression level between different groups (A) Differential expression levels of reference miR-16 ($^{\#}p > 0.05$). (B) The $\Delta$Ct values of miR-421 between different groups (****$p < 0.0001$). (C) The levels ($2^{-\Delta Ct}$) of miR-421 between different groups (****$p < 0.0001$). The $\Delta$Ct values denote the normalized Ct value by subtracting the Ct value of miR-16 from that of miR-421. The lower $\Delta$Ct value means the higher level of miR-421 expression. The Mann–Whitney $U$ test was used to determine statistical significance at the level of $p < 0.05$.

**Table 1** Differential expression levels of traditional tumor markers in plasma of GC, Pre and health controls.

| | GC | | Pre | | Pre vs NC | NC | | GC vs NC |
|---|---|---|---|---|---|---|---|---|
| | $n$ | mean ± SD | $n$ | mean ± SD | $p$ value | $n$ | mean ± SD | $p$ value |
| AFP | 85 | (26.14 ± 133.56) | 66 | (3.32 ± 1.72) | 0.0505 | 45 | (3.69 ± 1.25) | 0.3002 |
| CEA | 85 | (32.23 ± 128.76) | 66 | (1.82 ± 1.39) | 0.3122[a] | 45 | (2.08 ± 1.20) | 0.0011 |
| Ferritin | 80 | (281.08 ± 383.42) | 61 | (270.11 ± 302.39) | 0.1270[a] | 45 | (185.67 ± 237.46) | 0.9908 |
| CA125 | 68 | (59.22 ± 92.72) | 53 | (12.50 ± 7.78) | 0.9363 | 45 | (11.07 ± 4.05) | 0.0091 |
| CA153 | 67 | (13.44 ± 17.70) | 50 | (10.41 ± 5.45) | 0.0065[a] | 45 | (13.67 ± 5.83) | 0.0049 |
| CA199 | 83 | (107.03 ± 251.90) | 57 | (13.95 ± 12.37) | 0.1071 | 45 | (14.58 ± 8.22) | 0.0705 |
| CA211 | 62 | (16.15 ± 44.84) | 40 | (2.96 ± 1.40) | <0.0001 | 45 | (2.09 ± 0.48) | 0.0002 |
| CA242 | 66 | (19.17 ± 39.24) | 45 | (4.92 ± 4.8) | 0.2976[a] | 36 | (6.14 ± 5.48) | 0.1425 |
| CA50 | 77 | (38.01 ± 99.02) | 48 | (8.18 ± 5.97) | 0.0101[a] | 34 | (4.91 ± 4.46) | 0.0089 |
| CA724 | 75 | (15.12 ± 38.23) | 44 | (2.49 ± 2.87) | 0.2347[a] | 26 | (3.32 ± 2.59) | 0.2049 |

**Notes.**

[a]For these categorical variables, the $p$ value were calculated by Student's $t$-test. Mann–Whitney $U$ test was used to determine other categorical variables with statistical significance at the level of $p < 0.05$.

that miR-421 was significantly up-regulated in GC and Pre patients compared with the control group (all $p < 0.001$, Figs. 1B, 1C). The concentrations of CEA, CA125, CA153, CA211, and CA50 in the GC group were significantly higher than in the healthy controls (all $p < 0.05$, Table 1). Moreover, the concentration of CA211 and CA50 in the Pre group was significantly higher than the healthy controls (all $p < 0.05$, Table 1). Of note, compared with the precancerous group, the concentration of AFP and CA153 was significantly higher in the normal group (all $p < 0.05$, Table 1).

## miR-421 in plasma had higher diagnostic value as a non-invasive biomarker than traditional tumor markers for gastric cancer

To further evaluate the diagnostic value of miR-421 for GC, receiver-operating characteristic (ROC) curve analyses were performed (Fig. 2A and Table 2). According to the ROC curve, when the cut-off of 3.23, the optimal sensitivity and specificity of miR-421 were 96.67% and 95.56% respectively [AUC = 0.981 (0.942–0.997)]. To further

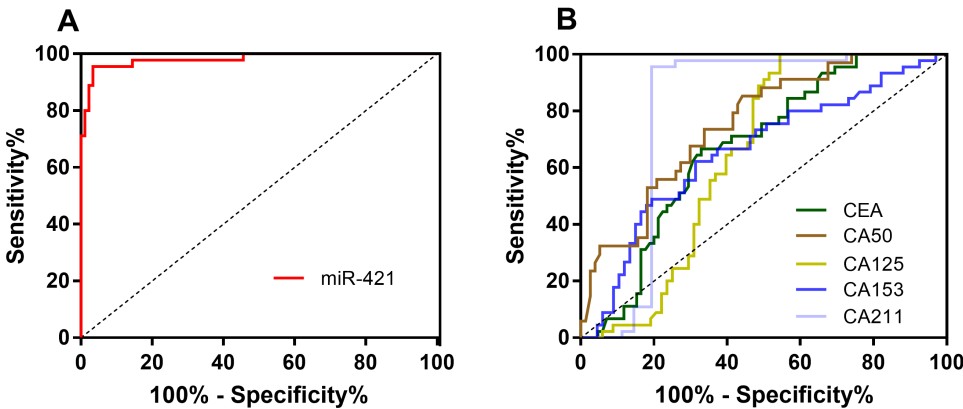

**Figure 2** The ROC curves of miR-421 and traditional tumor markers for GC. (A) The AUC of miR-421 was 0.981 ($p < 0.05$). (B) The AUC of CEA, CA50, CA125, CA153 and CA211 were 0.672, 0.754, 0.645, 0.656 and 0.799, respectively (all $p < 0.05$).

**Table 2** The diagnostic value of miR-421 and tumor markers for GC.

| | Cut-off | Sensitivity (%) | Specificity (%) | FPR (%) | FNR (%) | Accuracy (%) | Youden index (%) | AUC | p |
|---|---|---|---|---|---|---|---|---|---|
| miR-421 | 3.23 | 96.67 | 95.56 | 4.44 | 3.33 | 96.30 | 92.22 | 0.981(0.942–0.997) | <0.0001 |
| CEA | 1.94 | 67.06 | 66.67 | 33.33 | 32.94 | 66.93 | 33.73 | 0.672(0.584–0.752) | 0.0002 |
| CA125 | 18.75 | 45.59 | 100.00 | 0 | 54.41 | 67.26 | 45.59 | 0.645(0.549–0.732) | 0.0055 |
| CA153 | 11.31 | 68.66 | 62.22 | 37.78 | 31.34 | 66.07 | 30.88 | 0.656(0.561–0.743) | 0.0034 |
| CA211 | 2.2 | 80.65 | 95.65 | 4.35 | 19.35 | 86.96 | 76.30 | 0.799(0.711–0.871) | <0.0001 |
| CA50 | 7.96 | 55.84 | 85.29 | 14.71 | 44.16 | 64.86 | 41.14 | 0.754(0.663–0.831) | <0.0001 |

Notes.

AUC, Area under the curve; FPR, False positive rate; FNR, False negative rate.

Positive was defined as > cut-off value; Negative was defined as < cut-off value; The sensitivity, specificity, FPR, FNR, accuracy and Youden index were calculated based on the number of positive cases and negative cases.

**Table 3** The comparison of ROC curves between miR-421 and tumor markers.

| ROC curves comparison | CEA | | CA125 | | CA153 | | CA211 | | CA50 | |
|---|---|---|---|---|---|---|---|---|---|---|
| | Z | p | Z | p | Z | p | Z | p | Z | p |
| miR-421 | 2.418 | 0.0156 | 1.896 | 0.0580 | 2.021 | 0.0432 | 3.975 | 0.0001 | 3.343 | 0.0008 |

Notes.

ROC curves comparisons were analyzed by using MedCalc statistical software with statistical significance at the level of $p < 0.05$.

clarify the diagnostic efficiency of miR-421 in gastric cancer, the diagnostic values of traditional tumor markers were compared to those of miR-421. Figure 2B shows that the AUC, sensitivity, accuracy and Youden index of miR-421 were higher than all five conventional biomarkers, while the specificity was higher than CEA, CA153, and CA50. Table 3 shows the direct comparison of the ROC curves of circulating miR-421 with five conventional tumor markers. The diagnostic efficacy of miR-421 was significantly higher than CEA, CA153, CA211, and CA50 (all $p < 0.05$). In order to determine whether the

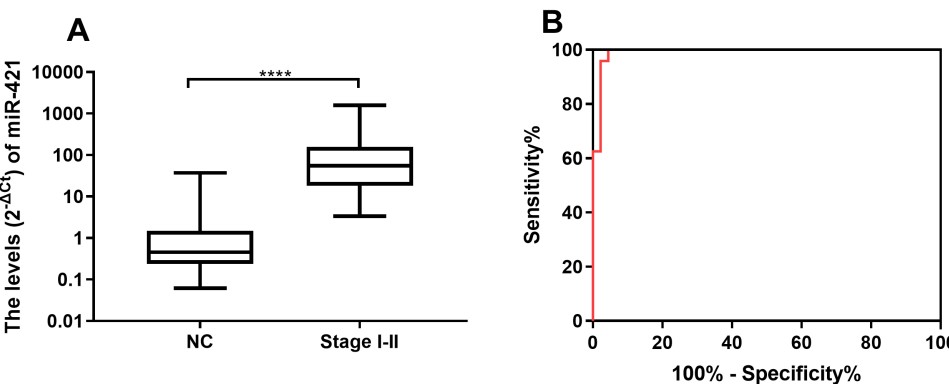

**Figure 3  MiR-421 could distinguish early stage gastric cancer patients from healthy controls.** (A) MiR-421 was up-regulated in early stages (TNM stage I–II) gastric cancer patients compared to healthy controls (****$p < 0.0001$). (B) The ROC analysis for detection of early-stages gastric cancer patients from health controls using miR-421.

plasma level of miR-421 has clinical value for GC diagnosis, the false positive rate (FPR) and false negative rate (FNR) and diagnosis efficiency were calculated. As shown in Table 2, the results indicate that the FNR of miR-421 was lower than all five conventional biomarkers, while the FPR was lower than CEA, CA153, and CA50, which indicated that miR-421 has a high diagnostic value for GC. These results indicate that the plasma miR-421 has greater diagnostic value for GC than five conventional tumor markers and can be used as a non-invasive diagnostic marker for gastric cancer.

### miR-421 in plasma could be used to distinguish early-stage GC patients from healthy controls

To evaluate the potential of the differentially expressed miR-421 to diagnose early-stage gastric cancer patients, we further analyzed the expression levels of miR-421 in GC patients at an early stage (TNM stages I–II). The results showed that the expression levels of circulating miR-421 in healthy controls were significantly lower than those in early-stage gastric cancer patients ($p < 0.0001$, Fig. 3A). The AUC values of miR-421 were 0.9907 (95% confidence interval (CI) = 0.931 to 1.0, sensitivity = 100% and specificity = 95.56%, Fig. 3B). These results indicate that miR-421 could be invoked as a novel biomarker for early diagnosis of gastric cancer.

### miR-421 had higher diagnostic value than traditional tumor markers for precancerous gastric lesions

Based on the verified difference of miR-421 between the Pre and NC group (Figs. 1B, 1C), ROC curves were constructed to further assess the role of miR-421 expression in the diagnosis of precancerous lesions. According to the different concentration of traditional tumor markers between the Pre group and the NC group (Figs. 4A, 4B), their diagnostic values with miR-421 were further compared. ROC analyses showed that up-regulation of plasma miR-421 could discriminate Pre patients well from control subjects, with an AUC value of 0.873 (sensitivity = 66.29%, specificity = 95.56%, cutoff value: 3.23) (Fig. 4C).

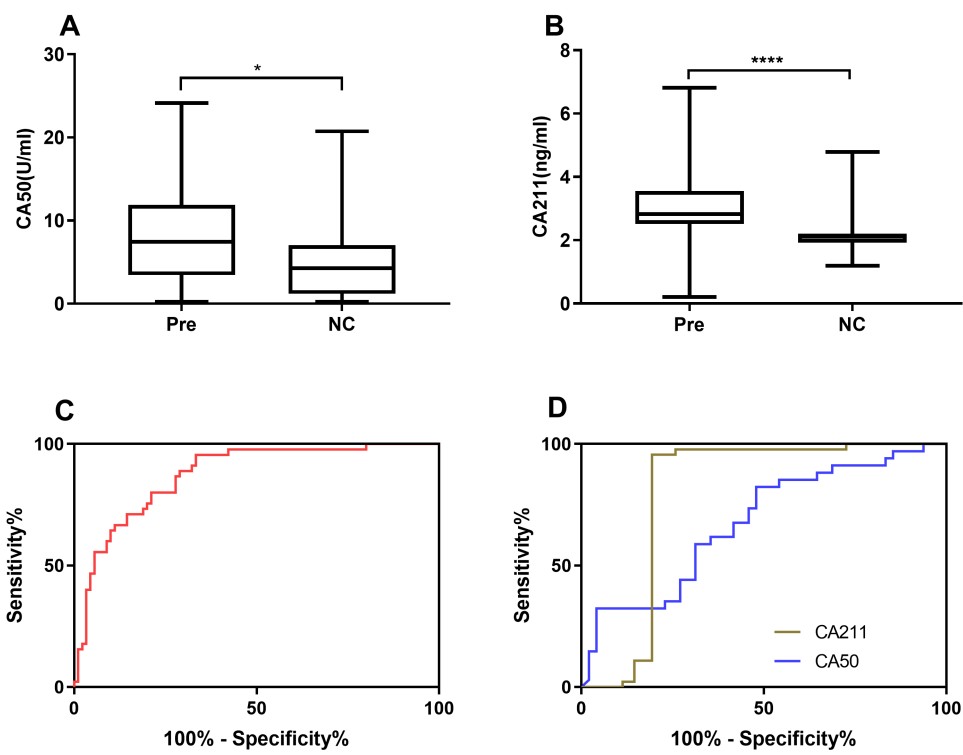

**Figure 4** **miR-421 had higher diagnostic value than traditional tumor markers for precancerous patients.** Differential expression levels of CA50 (A) and CA211 (B) between precancerous patients and healthy controls (*$p < 0.01$, ****$p < 0.0001$). The ROC analysis for detection of precancerous using miR-421 (C) and traditional tumor markers CA50 and CA211 (D).

**Table 4** The diagnostic value of miR-421 and tumor markers for Pre.

|  | Cut-off | Sensitivity (%) | Specificity (%) | FPR (%) | FNR (%) | Accuracy (%) | Youden index (%) | AUC | *p* |
|---|---|---|---|---|---|---|---|---|---|
| miR-421 | 3.23 | 66.29 | 95.56 | 4.44 | 33.71 | 76.37 | 61.85 | 0.872 (0.804–0.924) | <0.0001 |
| CA153 | 11.81 | 72.00 | 60.00 | 40.00 | 28.00 | 66.32 | 32.00 | 0.679 (0.576–0.772) | 0.0012 |
| CA211 | 2.2 | 85.00 | 95.56 | 4.44 | 15.00 | 90.56 | 80.56 | 0.844 (0.750–0.914) | <0.0001 |
| CA50 | 7.26 | 52.08 | 82.35 | 17.65 | 47.92 | 64.63 | 34.44 | 0.677 (0.564–0.776) | 0.0034 |

**Notes.**

The sensitivity, specificity, FPR, FNR, accuracy and Youden index were calculated based on the number of positive cases and negative cases. AUC, FPR and FNR, Positive and Negative as stated in Table 2.

Furthermore, as shown in Table 4 and Figs. 4C & 4D, the AUC of miR-421 was higher than CA153, CA211, and CA50, the sensitivity was higher than CA50, while the specificity and Youden index were higher than CA153, and CA50. Moreover, the FNR of miR-421 was lower than CA50, while the FPR was lower than CA153, and CA50. According to the comparison of the ROC curves, the diagnostic efficacy of miR-421 was considerably higher than CA153, CA211 and CA50 (all $p < 0.05$, Table 5). These results indicate that miR-421 has high diagnostic value for precancerous patients.

**Table 5 The comparison of ROC curves among miR-421 and tumor markers for Pre.**

| ROC curves comparison | CA153 | | CA211 | | CA50 | |
|---|---|---|---|---|---|---|
| | $Z$ | $p$ | $Z$ | $p$ | $Z$ | $p$ |
| miR-421 | 2.211 | 0.027 | 4.414 | <0.0001 | 1.997 | 0.0458 |

**Notes.**
ROC curves comparisons were analyzed by using MedCalc statistical software with statistical significance at the level of $p < 0.05$.

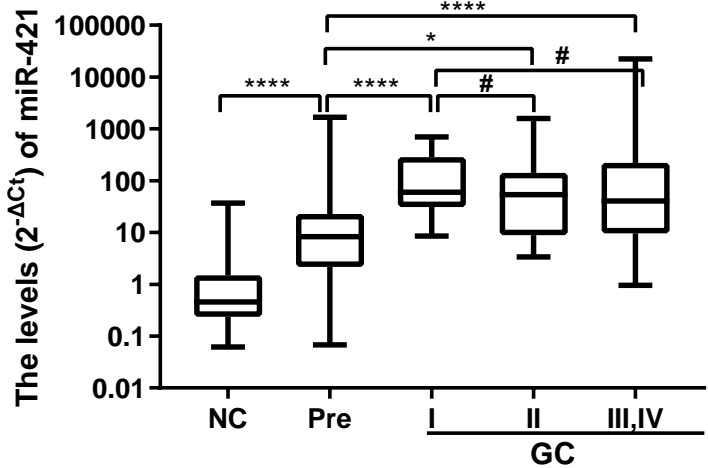

**Figure 5 MiR-421 expression levels in plasma of GC cases with different TNM stages, PLGC cases and NC controls.** Mann–Whitney $U$ test was used to determine statistical significance, $p < 0.05$ was accepted as statistically significant. $^*p < 0.05$, $^{****}p < 0.0001$, $^\#p > 0.05$.

## Up-regulation of miR-421 occurred initially in the precancerous diseases as well as the early GC

We found that up-regulation of miR-421 not only distinguishes early-stage gastric cancer well, but can also well distinguish precancerous lesions from normal controls. However, it is not clear at what stage miR-421 begins to be up-regulated throughout the development of GC. Therefore, expressions of miR-421 in precancerous disease, and different clinical stages (I, II, III and IV) were further analyzed and compared. The results showed that the expressions of miR-421 were statistically different between the Pre cases and GC stages I, II, III and IV ($p < 0.05$, Fig. 5). However, there was no difference of miR-421 expression between the GC stages II, III and IV and GC cases stage I. Moreover, compared to the expression in Pre cases, the biomarker was gradually up-regulated from the NC to the Pre and stage I cases, and finally, the maximum expression value appeared in stage I cases (Table 6). The marked up-regulation of miR-421 occurred initially in precancerous disease as well as early-stage GC.

## Association of miR-421 with clinicopathological features

In order to determine whether miR-421 levels in plasma were correlated with clinicopathological features in patients with GC and precancerous lesions, the clinical

**Table 6  The levels $(2^{-\Delta Ct})$ of miR-421 between different groups.**

|  | $n$ | miR-421 in plasma | Sd | $^aP$ value | $^bP$ value |
|---|---|---|---|---|---|
| NC | 45 | 0.45(0.16,37.0) | 5.50 | | |
| Pre | 89 | 7.57(0.06,1674.25) | 229.36 | | |
| I | 15 | 59.97(8.54,702.0) | 187.37 | <0.0001 | |
| II | 9 | 53.60(3.3,1577.80) | 510.76 | 0.0277 | 0.3786 |
| III and IV | 63 | 40.72(0.95,22511.89) | 3,020.8 | <0.0001 | 0.3325 |

Notes.

$^a P$ represents the comparison of Pre between GC with I, II, III and VI respectively.

$^b P$ represents the comparison of I between GC with II, III and respectively.

data of 89 Pre and 90 GC-proven cases were collected and analyzed. The high and low expression groups of each parameter were calculated by the median of the relative quantity of miR-421. Correlation analysis showed no statistically significant association of miR-421 expression with GC patients' age ($p = 0.0736$), gender ($p = 0.3265$), TNM stage ($p = 0.3531$), differentiation degree ($p = 0.5105$), and histological type ($p = 0.5079$). Similarly, we found that there were no apparent correlations of miR-421 expression with Pre patients' age ($p = 0.5896$), gender ($p = 0.5934$), histological type ($p = 0.6939$), and *Helicobacter pylori* infection status ($p = 0.3095$) (Table 7).

# DISCUSSION

To date, there is no reliable non-invasive blood-based biomarker for the detection of precancerous lesions and early-stage gastric cancer. We studied the diagnostic value of miR-421 in plasma in precancerous gastric lesions and GC patients. We found that the plasma expression levels of miR-421 were statistically significantly higher in early-stage GC and precancerous gastric lesion patients than in healthy controls. ROC analyses demonstrated that miR-421 has high sensitivity and specificity in discriminating early-stage GC and precancerous gastric lesion patients from healthy controls. Our results indicated that miR-421 in plasma presented statistically significant diagnostic efficacy in detecting early GC as well as precancerous gastric lesions patients.

High-throughput analysis showed that miR-421 is localized on human chromosome1p34.23, and is one prominent member of the miR-200 family (*Liu et al., 2015*). *Liu et al. (2015)* reported that miR-421 elevation in GC tissue (*Zhao et al., 2015*) and gastric juice (*Zhang et al., 2012b*) probably had implications in GC diagnosis. These results are consistent with our findings that miR-421 was highly expressed in the plasma of gastric cancer patients might have the potential diagnostic value in cancer. However, the fact that these researches are based on invasive techniques that patients may not be willing to accept is bound to affect their promotion. Moreover, our study showed that miR-421 could achieve a satisfactory diagnostic efficiency in distinguishing GC patients from healthy controls with an AUC of 0.981 (sensitivity = 96.67% and specificity = 95.56%), higher than miR-21, miR-20a and miR-378 tumor markers in previous studies (*Liu et al., 2012*; *Mirzaei et al., 2016*). In our study, we compared its diagnostic efficiency with that of traditional markers, further clarifying its diagnostic value as a new tumor marker.

**Table 7** The relationships between the expression levels of miR-421 ($2^{-\Delta Ct}$) in plasma and clinicopathological factors of patients with precancerous gastric lesions and gastric cancer.

| Characteristics | Case No. | miR-421 level ($2^{-\Delta Ct}$) | | $\chi 2$ test | P value |
|---|---|---|---|---|---|
| | | High | Low | | |
| Age(years) | | | | | |
| <60 | 30 | 11 | 19 | 3.2 | 0.0736 |
| ≥60 | 60 | 34 | 26 | | |
| Gender | | | | | |
| Male | 68 | 32 | 36 | 0.9626 | 0.3265 |
| Female | 22 | 13 | 9 | | |
| TNM stage | | | | | |
| I | 16 | 11 | 5 | 3.261 | 0.3531 |
| II | 9 | 5 | | | |
| III | 23 | 12 | 11 | | |
| VI | 40 | 17 | 23 | | |
| Diferentiation degree | | | | | |
| High and moderate | 25 | 11 | 14 | 0.433 | 0.5105 |
| Poor | 46 | 24 | 22 | | |
| Histological type | | | | | |
| Adenocarcinoma (A) | 65 | 31 | 34 | 2.324 | 0.5079 |
| Mucinous carcinoma(M) | 2 | 0 | 2 | | |
| Signet ring cell carcinoma(S) | 5 | 3 | 2 | | |
| A with S | 11 | 6 | 5 | | |
| **Pre** Total cases | 89 | 45 | 44 | | |
| Age(years) | | | | | |
| <60 | 41 | 22 | 19 | 0.2916 | 0.5892 |
| ≥60 | 48 | 23 | 25 | | |
| Gender | | | | | |
| Male | 46 | 22 | 24 | 0.2851 | 0.5934 |
| Female | 43 | 23 | 20 | | |
| Histological type | | | | | |
| Intestinal metaplasia | 82 | 41 | 41 | 0.1316 | 0.6939 |
| Atypical hyperplasia and other type | 7 | 4 | 3 | 0.1316 | 0.6939 |
| HP infection status | | | | | |
| HP (−) | 38 | 20 | 18 | 1.033 | 0.3095 |
| HP (+) | 16 | 6 | 10 | 1.033 | 0.3095 |

It is particularly important to improve the early diagnosis and prompt treatment of patients with gastric cancer. Researchers have illuminated the potential roles of several miRNAs in the early detection of cancers, such as lung carcinoma (*Pan et al., 2018*), colorectal neoplasm (*Liu et al., 2018*), and early-stage breast carcinoma (*An et al., 2018*). However, to date only a few miRNAs have been found effective as non-invasive biomarkers for the early detection of GC. Although the study such as the five-miRNA panel presented a high diagnostic value for the early stage of GC (*Liu et al., 2012*), the study did not include gastric precancerous lesions, and the subtle changes and performances of the miRNAs were

not further elaborated. The miR-17-92 cluster was considered as a circulating biomarker for early detection of GC, but there is no ROC data to support miRNA recognition of early stage cancer (*Li et al., 2017*). In our study, based on the detailed TNM stage characteristics of the GC sample, we explored the diagnostic value of plasma miR-421 in early gastric cancer. The results showed that the expression level of miR-421 could well distinguish early gastric cancer from normal controls, suggesting that miR-421 may have significant clinical value in the early diagnosis of gastric cancer.

Precancerous changes in the stomach can increase the risk of gastric cancer six-fold. Previous studies have reported that precancerous lesions are regulated by miRNAs (*Li et al., 2014*; *Rotkrua et al., 2011*).

Due to the absence of clinical manifestations of gastrointestinal metaplasia and atypical hyperplasia, there are few means for clinical non-invasive detection. Several circulating miRNAs, especially microRNA-196a (*Chen et al., 2018*), miR-16-5p, and miR-19b-3p (*Zhang et al., 2015*), have been screened out for early markers of gastric cancer and precancerous lesions and of GC progression. However, the comparative value of these miRNAs and traditional tumor markers in the diagnosis of gastric cancer and precancerous lesions has not been clarified, and an optimal circulating miRNA biomarker should have better specificity and sensitivity than traditional tumor markers. In the current study, we found that plasma miR-421 could well distinguish precancerous lesions of gastric cancer patients from healthy controls with an AUC of 0.872 (sensitivity = 66.29% and specificity = 95.56%). Furthermore, the diagnostic efficacy of miR-421 was markedly higher than traditional tumor markers, such as CA153, CA211, and CA50.

The occurrence and development of gastric cancer is a multi-stage process, and miR-421 may be associated with a certain stage or the whole process. In this study, we further analyzed the trend of the up-regulation of miR-421 with the progression of GC. Results showed that the expression of plasma miR-421 in GC cases at stage I is no different from stages II, II, and IV, but plasma miR-421 in precancerous cases can be distinctly distinguished from GC at each stage, indicating that miR-421 occurred initially in precancerous lesions as well as early GC. Therefore, we speculate that the up-regulation in the miR-421 may be triggered by risk factors, especially chronic inflammation in the premalignant stomach, suggesting that a detailed investigation is needed.

## CONCLUSIONS

Our findings demonstrate that plasma miR-421 may be a potential marker for precancerous lesions and early gastric cancer. This study provides us with a new insight into the role of miR-421 in gastric cancer. These results could pave the way for the development of a novel biomarker for the early diagnosis of gastric cancer. Using this biomarker may allow a reliable indicator of gastric cancer at a much lower cost than methods currently available. The limitations of this study are that our normal control samples were relatively small, and most cancer samples were at an advanced stage. The exact use of miR-421 as an early

plasma biomarker in clinical diagnosis of precancerous lesions and gastric cancer should be further explored in studies that use larger samples and in clinical trials.

### Funding

This study was funded by the National Natural Science Foundation of China [Grant No. 81760501], the Natural Science Foundation of the Guangxi Zhuang Autonomous Region (Grant No. 2017GXNSFAA198041), Liuzhou scientific research and technological development programs (Grant No. 2014JC010), the Self-Funded Research Project of Guangxi Zhuang Autonomous Region Health and Family Planning Commission (Grant No. Z2014613). The funders had no role in study design, data collection and analysis, decision to publish, or preparation of the manuscript.

### Grant Disclosures

The following grant information was disclosed by the authors:
National Natural Science Foundation of China: 81760501.
The Natural Science Foundation: 2017GXNSFAA198041.
Liuzhou Scientific Research and Technological Development Programs: 2014JC010.
Guangxi Zhuang Autonomous Region Health and Family Planning Commission: Z2014613.

### Competing Interests

The authors declare there are no competing interests.

### Author Contributions

- Jianlin Chen conceived and designed the experiments, performed the experiments, analyzed the data, contributed reagents/materials/analysis tools, prepared figures and/or tables, authored or reviewed drafts of the paper, approved the final draft.
- Lihua Wu conceived and designed the experiments, performed the experiments, contributed reagents/materials/analysis tools, authored or reviewed drafts of the paper, approved the final draft.
- Yifan Sun, Qi Yin, Xianhua Chen, Siqun Liang, Qingyan Meng, Haihua Long, Fangying Li, Changjun Luo, and Xiaorong Xiao contributed reagents/materials/analysis tools, approved the final draft.

### Ethics

The following information was supplied relating to ethical approvals (i.e., approving body and any reference numbers):

This study was approved by the Ethics Committee of the Affiliated Liutie Central Hospital of Guangxi Medical University and written informed consent was issued by all study participants.

## Data Availability

Raw data is available in Supplemental Files.

## Supplemental Information

Supplemental information for this article can be found online at http://dx.doi.org/10.7717/peerj.7002#supplemental-information.

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
