# Peer review of "Mir-421 in plasma as a potential diagnostic biomarker for precancerous gastric lesions and early gastric cancer"

_PeerJ, doi:10.7717/peerj.7002_

## Round 0.1 · original submission · Major Revisions

Please, follow the reviewers suggestions to improve your manuscript.

Reviewer 1 ·

Basic reporting

Authors should check their references. there are so many reviews as references which is not suitable. original authors must be mentioned rather than reviewers.

miRNA expression details should be given.

Experimental design

no comment

Validity of the findings

Raw Cts' extended over 30 cycles which is not acceptable for a valid PCR result. RNA quality and quantity (semi) and the PCR procedures, like if there is any preamplification step carried out or not should be given in details.

Additional comments

Authors referred some reviews that actually don’t carry out the study of relevant findings. If authors want to add a review to the article they should add it as an advisory or as a recommendation not as a reference. So Authors must check the references and find the original articles instead of reviews. Two of them are;
a) Leung et al 2008
b) Park & Kim 2015


Reference for miRNA definition is not suitable as every researcher in this field knows that Victor Ambros (2001) must be referred for that definition.

The sentence starting in line 46 does not need a reference and moreover reference given for the sentence is not suitable.

The sentence starting with at present in line 49 does not need a reference rather it is a conclusion sentence completing the previous sentences. Also both references are reviews.

In line 101 Union of “International Control Cancer” should be “Union of International Cancer Control”.
For delta delta Ct method Pan et al is not the reference (Pan et al, 2018) to be used as the mentioned author didn’t generated the formula.

Authors should specify the brand with catalog number of the Helicobacter Pylori assay.

Did authors measured RNA quantity after RNA isolation (via nanodrop ie)?

Authors should give the details of Quantitaion of miRNAs. And specify if there is any preamp step, cDNA step and which chemistry used for monitoring PCR products like sybergreen taqman scorpion probes.

What was the threshold for non-specific PCR products. In figure 1a GC miR-16 raw Cts’ showed a wide range of threshold and looks like authors didn’t rule out any PCR products over any threshold (SD extended over 30 cycles). Authors should check real time PCR graphs if there are any false positive results. and there should be a limit for Ct values for being accepted as it worked.

Authors used miR-16 as a reference for comparison. In figure 1a raw Ct of GC patients are wider. Did authors checked if those samples have any problems due to any procedure problems or maybe an overlooked patient with any contributing health problem.

Reviewer 2 ·

Basic reporting

Despite the goal and the message of the work is clear, english must be absolutely revised.
Furthermore, the authors should improve introduction to give sufficient field background to the reader. Several papers investigated the expression of circulating miRNAs as early markers of gastric cancer and precancerous lesions and of GC progression (Chen TH, et al. Circulating microRNA-196a is an early gastric cancer biomarker. Oncotarget. 2017 Dec 7;9(12):10317-10323. doi:10.18632/oncotarget.23126; Zhang J et al. Circulating MiR-16-5p and MiR-19b-3p as Two Novel Potential Biomarkers to Indicate Progression of Gastric Cancer. Theranostics. 2015 Apr 5;5(7):733-45. doi: 10.7150/thno.10305. eCollection 2015; Wang XW et al. MicroRNA network analysis identifies key microRNAs and genes associated with precancerous lesions of gastric cancer. Genet Mol Res. 2014 Oct 27;13(4):8695-703. doi: 10.4238/2014.October.27.10). Importantly, Zhang and co-workers, initially screened out differentially expressed miRNAs by genome-wide miRNA profiling microarrays and validated some of them in a larger cohort of patients. Authors should quote this article and discuss the results obtained on the basis of those already existing in the literature.

Experimental design

The research question is relevant and meaningful, however the authors should keep in mind that scientific works already exist in which the expression of some microRNAs has been analyzed as markers of early diagnosis and precancerous lesions. As these miRNAs are different from miR-421, the authors should validate and compare their level of expression, sensitivity and accuracy with that of miR-421 and discuss these results.
Methods are described with sufficient information.

Validity of the findings

Data are robust and statistically sound. Patient cohort is numerically good to reach the conclusion. Conclusion is well stated and linked to the riginal research question.

Additional comments

The work is well designed and structured, however it should be improved both in the introduction and in the results section: The authors should consider the other works in the same field of research and analyse the expression of the miRNAs already published as potential markers of early diagnosis of GS and precancerous lesions in order to compare thei expression with miR-421. These results should also discussed.

---

## Round 0.2 · Minor Revisions

A few more revisions should be performed:

1. miR-16 is not an internal control but the reference miRNA.
2. In figure 1B, N group should be NC
3. It seems from figure 1C that the miR-421 has been reported as 2-delta delta Ct (fold change vs control NC), but in materials and methods it is reported as 2-delta Ct (fold change vs miR-16), please check how the relative expression has been calculated.
4. change the sentence "Samples with Ct values greater than 30 will be excluded" with "Samples with Ct values greater than 30 were excluded"

Reviewer 2 ·

Basic reporting

No comment

Experimental design

No comment

Validity of the findings

No comment

Additional comments

The authors reviewed the work according to my comments.
In particular they revised the form, added some important references and expanded the introduction to make the experimental part clearer.
I think the work is now suitable for publication

---

## Round 0.3 · accepted · Accept

The revisions have been performed following reviewers suggestions.

#